# Beyond Avoiding Hemiplegia after Glioma Surgery: The Need to Map Complex Movement in Awake Patient to Preserve Conation

**DOI:** 10.3390/cancers15051528

**Published:** 2023-02-28

**Authors:** Fabien Rech, Hugues Duffau

**Affiliations:** 1Department of Neurosurgery, CHRU de Nancy, Université de Lorraine, F-54000 Nancy, France; 2Le Centre de Recherche en Automatique de Nancy, Le Centre National de la Recherche Scientifique, Université de Lorraine, F-54000 Nancy, France; 3Department of Neurosurgery, Gui de Chauliac Hospital, Montpellier University Medical Center, F-34295 Montpellier, France; 4Team ‘Plasticity of Central Nervous System, Stem Cells and Glial Tumours’, INSERM U1191, Institute of Genomics of Montpellier, University of Montpellier, F-34295 Montpellier, France

**Keywords:** awake surgery, glioma, motor function, conation, movement control

## Abstract

**Simple Summary:**

Glioma surgery relies on the ability to perform a large extent of resection while preserving the patient’s quality of life, especially regarding complex movement. Our aim is to show how the concept of motor function has evolved based upon an increased knowledge of its neural foundation in neurosciences and how this understanding has shed light on possible disturbances of conation. Postoperative troubles can be avoided thanks to the implementation of adapted intraoperative tasks during awake surgery, from the basic muscle contraction to prevent hemiplegia (first level), to active movement to avoid fine motor disturbances (second level) and even to multitasking to preserve an intact movement volition.

**Abstract:**

Improving the onco-functional balance has always been a challenge in glioma surgery, especially regarding motor function. Given the importance of conation (i.e., the willingness which leads to action) in patient’s quality of life, we propose here to review the evolution of its intraoperative assessment through a reminder of the increasing knowledge of its neural foundations—based upon a meta-networking organization at three levels. Historical preservation of the primary motor cortex and pyramidal pathway (first level), which was mostly dedicated to avoid hemiplegia, has nonetheless shown its limits to prevent the occurrence of long-term deficits regarding complex movement. Then, preservation of the movement control network (second level) has permitted to prevent such more subtle (but possibly disabling) deficits thanks to intraoperative mapping with direct electrostimulations in awake conditions. Finally, integrating movement control in a multitasking evaluation during awake surgery (third level) enabled to preserve movement volition in its highest and finest level according to patients’ specific demands (e.g., to play instrument or to perform sports). Understanding these three levels of conation and its underlying cortico-subcortical neural basis is therefore critical to propose an individualized surgical strategy centered on patient’s choice: this implies an increasingly use of awake mapping and cognitive monitoring regardless of the involved hemisphere. Moreover, this also pleads for a finer and systematic assessment of conation before, during and after glioma surgery as well as for a stronger integration of fundamental neurosciences into clinical practice.

## 1. Introduction

One major concern about brain surgery, especially for glioma, has always been the preservation of motor function given its importance in patient’s quality of life as well as the frequency of glioma invading premotor and motors structures. Notably, the motor system has extensively been studied since the seminal work of Hughlings Jackson [1]. In animals, lesion studies, electrical stimulations and tracer injections have been used to understand the function and somatotopy of premotor and motor areas [2,3,4,5,6,7] as well as their interactions in the context of an hodotopic system rather than a serial one [8,9]. Functional imaging in humans [10,11,12,13] and non-invasive brain stimulation techniques [14] have been helpful to define the framework of motor control in healthy volunteers.

Surprisingly, during the same period, the protocol of motor mapping has not actually changed since its first description, by Penfield, of the motor homunculus in awake patients [15]. The main objective of motor mapping has predominantly consisted of looking for the primary motor cortex (M1) and the pyramidal tract to avoid hemiplegia or hemiparesis via the use of direct electrostimulations (DES). To fulfill such an objective, awake brain surgery has even been frequently deprecated to the benefit of EMG/MEP under general anesthesia. Nevertheless, several studies have shown that permanent movement disturbances might occur after surgery despite preservation of the primary motor cortico-subcortical structures [16,17,18,19,20]. Such reports have given rise to the emergence of new protocols to assess the motor function more precisely [21,22] but also beyond, to preserve across-networks dynamics in which the motor network contributes [23]. We propose here to review how the motor function should be integrated in the framework of conation (i.e., the willingness which leads to action) to better consider patient expectations regarding their quality of life after brain surgery [24]. To this end, we reviewed the history of DES of the motor system and showed how intraoperative “motor” mapping with cognitive monitoring is now used to preserve the conation far beyond the primary motor structures.

## 2. First Level: Muscle Contraction and Motor Output

As mentioned above, the protocol of motor mapping first consisted of generating muscle contraction or involuntary dystonic movement thanks to DES in awake conditions, also called positive motor responses (PMR) [15]. This is helpful to identify M1, rostrally to the central sulcus and at the subcortical level, the pyramidal tract, projecting to the spinal cord (Figure 1).

Because such identification requires only a contraction, mapping has also been proposed by using EMG/MEP under general anesthesia to identify subcomponents of M1 and enables safe resection inside the primary motor cortex [25]. This kind of approach might be proposed when no cognitive task needs to be performed, for example, when a lesion invades only the primary motor structures or when the patient presents important preoperative motor impairment leading to the impossibility to assess movement in awake conditions as we will see further [26]. In fact, such strategy for tumor resection consists of identifying the output of an unimodal network, also theorized recently as a first level of neural disruption with DES [27]. This attitude is adapted to preserve one of the last relays of the system, without which there is no function because there is no motor output.

From an anatomical point of view, several strategies can be identified depending on tumor location. If the tumor invades the premotor area, mapping will identify positive motor sites caudally, and then the approach will be to perform the tumor resection rostro-caudally. During the resection, DES are also performed at the subcortical level to identify the pyramidal pathway and to stop the resection posteriorly into the contact of the cortico-spinal structures. A similar strategy can be used for tumor invading the retrocentral gyrus. In this situation, resection will be performed caudo-rostrally up to the identification of the primary motor structures as the anterior limit, if the somatosensory structures have been reorganized in reaction to the tumor growth. Another approach consists of going directly through the precentral gyrus when the tumor is located in M1, identifying PMR around the core of the tumor thanks to neuroplastic mechanisms induced by the glioma [28]. Usually, this leads to quickly identifying the functional boundaries since brain reshaping is often spatially limited to perilesional primary motor structures [29]. More in the depth, it is also possible to detect the pyramidal pathway during resection of insular tumor, near the posterior limb of the internal capsule [30].

However, although the removal of premotor structures has been considered safe for a long time, many studies have reported some degrees of permanent disturbances [16,17,18,19,20,31]. One major topic after surgery in premotor area has been the occurrence of the supplementary motor area (SMA) syndrome [31]. It consists of the occurrence of a contralateral hemicorporeal akinesia, possibly associated with a mutism in the dominant hemisphere for language. These symptoms tend to recover spontaneously, excepted for fine movements skills, bimanual coordination, sometimes with the onset of abnormal movements [16,31,32]. This recovery of the SMA syndrome, even if incomplete, is made possible thanks to postlesional brain plasticity processes, in particular, based upon the recruitment of the contralateral SMA [19,33,34]. However, long-term deficits might persist despite an effective mapping of MEP under general anesthesia, especially in the SMA, the anterior cingulate gyrus and the dorsal premotor cortex [35]. These permanent consequences are understandable because SMA is considered crucial for movement selection and execution [7,36], planning complex sequence of movement [37], generating self-initiated movement [38,39], managing conflict between potential actions [32] and in the temporal organization of multiple movements [40,41,42]. In addition, a finest postoperative assessment of patient who underwent surgery for a glioma located in the ventral parieto-frontal region has also shown a high proportion of ideomotor apraxia [22]. Such a region, whose function is to match the hand shape to an object shape, belongs to the grasping network, connecting the inferior parietal lobule and the intraparietal sulcus to the ventral premotor cortex (vPM) via the superior longitudinal fascicle (SLF) III [43,44]. Taken together, these results support the necessity for improvement of the intraoperative mapping from a low-level muscle contraction to a higher level of movement coordination to preserve quality of life.

## 3. Second Level: Movement Coordination and Control

In this spirit, it has been proposed to use DES to map a second level of motor function, i.e., coordination and control, by disrupting the ongoing movement of the contralateral upper limb (possibly lower limb) in awake patient [27]. Such observation has already been described by Penfield [45], but it was Lüders who formalized the concept of “negative motor response” (NMR), which corresponds to a complete arrest of the ongoing movement without loss of tonus or consciousness [46]. This phenomenon has then been reported in several studies using DES on premotor and motor areas, although its significance has not been perfectly understood [47]. Controversial hypotheses about the role of negative motor areas (NMA) (namely, areas whose stimulation elicit NMR) and especially about the actual meaning of the negative motor phenomenon have been evoked without initially leading to practical use [36,47,48,49]. Indeed, studies with resection of NMA reported only transient deficit with a “recovery considered as complete” in a few hours/days at a standard neurological examination [49,50]. However, this view should be balanced by the fact that plastic mechanisms following NMA resection can occur on the condition that a part of the cortex as well as the converging subcortical fibers are not damaged, as already described for M1 [28,51]. Moreover, when a more precise assessment of the motor function is achieved, the classical sites of NMA are those considered as responsible for long-term deficit after resection under general anesthesia with MEP [35,52,53]. Interestingly, Vigano et al. have recently reported that NMR might present a different EMG pattern, and thus, it is likely rely on different neural networks, therefore possibly explaining such controversies [54].

Considering these findings, NMR have also been searched in the white matter because DES are particularly effective at a axonal level to avoid performing large subcortical disconnection in less plastic structures [29]. Thanks to this method, NMR have been identified in the white matter beneath the premotor regions: importantly, preservation of such structures prevents the occurrence of postoperative SMA syndrome as well as permanent disturbances in subtle movement control [55,56]. From this time, it has been possible to dramatically improve the protocol of intraoperative mapping and to achieve more tailor-made resections by understanding better the cortico-subcortical organization of the motor control network [57] (Figure 2).

Concretely, the patient is asked to perform an intraoperative movement task while DES are applied on the brain to identify the cortical and subcortical structures involved in motor control. Two approaches are routinely used in the operating room. The first is to ask the patient to perform alternative flexion and extension of the contralateral (or bilateral) upper limb(s) at approximately 0.5 Hz (1 cycle of flexion extension every 2 s) with a simultaneous counting or naming task, and to look for movement slowdown/arrest and/or speech arrest during DES [52,58]. The second is to use an intraoperative hand manipulation task tool to assess more finely the sensorimotor integration and dexterity [22].

During surgery, at the cortical level, it is possible to identify NMA mainly over the precentral gyrus, on the dorsal premotor cortex (dPM) and ventral premotor cortex (vPM) [52]. NMA are distributed in three main locations for the upper limb and two locations for speech, overlapping each other. Interestingly, such segregation is consistent with previous data gained via functional and structural imaging of the precentral gyrus, which showed a dorso-ventral gradient of four components, each one being responsible of different aspects of hand and upper limb control [59]. These findings challenge the classical view of the motor homunculus, in favor of a more action driven organization, or ethological map of action [15,60,61]. Even if NMR can be elicited on the medial wall of the hemisphere, no precise areas have yet been identified [53]. This is likely because there is a continuum between the pre-SMA, which is connected to prefrontal areas and dedicated to cognitive process, and the SMA-proper, which is more connected to motor areas (M1, dPM and vPM) and dedicated to motor processes [62,63].

Usually, preserving movement coordination in frontal regions consists of removing glioma located in the superior frontal gyrus (SFG), middle frontal gyrus (MFG) and/or inferior frontal gyrus (IFG). In this configuration, NMA over the lateral part of the hemisphere should represent the posterior limit of the resection since they are commonly located posteriorly to the precentral sulcus [52]. In addition, because diffuse glioma migrates mainly along the white matter pathway, the underlying connectivity must also be detected by DES and preserved. The SMA, involved in movement coordination and initiation, is densely connected with the dPM and vPM as well as with the IFG and the subthalamic nucleus to handle inhibition [62,64,65,66,67,68]. Connections with the dPM and vPM are going more posteriorly than those with the IFG which are going laterally and more anteriorly. They rely on the frontal aslant tract (FAT), a large white matter pathway of the frontal lobe connecting the SFG to the IFG and the vPM (Figure 3) [65,68,69].

One major objective is to avoid removing part of the FAT that still connects the SMA to the IFG despite glioma growth, specifically on the left side. Indeed, it has been shown that resection up to the left FAT generates disturbances in verbal fluency and speech initiation [68]. Consequently, it is crucial to perform a speech task during resection in such location (in addition to limb movement) and to be very attentive to slowdown of speech or difficulty of initiation during DES. Moreover, dPM, vPM and SMA are connected to the basal ganglia by means of the fronto-striatal tract (FST), an assembly of white matter fibers descending from these premotor areas to the striatum [44]. Importantly, resection up to the FST might lead to troubles not only in verbal fluency but also in motor initiation in both hemispheres [68]. At the level of the SMA, the FAT and FST are intermingling with each other, so it can be difficult to know which fascicle is stimulated, sometimes both—except by getting closer to the head of the caudate in which only the FST runs [44,68,69]. During resection, when approaching the functional part of the complex FAT/FST at the level of the SFG, DES might induce NMR of the lower limb medially and posteriorly and then NMR of the upper limb more laterally. It is also possible to identify NMR involving bilateral upper limbs. Finally, NMR of face/speech can be generated more laterally, corresponding here to the FAT which is running slightly more antero-laterally to the FST. Distribution of NMR sites in the white matter of the premotor area is therefore somatotopic [21] and generally corresponds to the somatotopy of the SMA, namely, from rostral to caudal: first with speech in the dominant hemisphere for language, then face, upper limb and finally lower limb [6,70,71]. This subcortical distribution of white matter bundles also corresponds well to the distribution of NMA over the lateral premotor areas and, despite variability across subjects, is reliably identified at the individual level [21,52]. It is noteworthy that in traditional surgery classically achieved in asleep patients, all these NMAs and corresponding subcortical fibers are removed during the rostro-caudal resection of tumor up to the pyramidal tract (as mentioned in the previous section). This explains why permanent complex movement deficit may occur by mapping only the primary motor system [56].

Interestingly, considering these lateral premotor areas, the cortical dorso-ventral gradient over the dPM and vPM previously detailed corresponds to the functional specialization of upper limb movement. Indeed, the dPM, which is mainly connected to the SMA, M1, the vPM and the superior parietal lobule [8,72], plays a role in reaching by converting spatial coordinates of a target obtained from the superior parietal lobule into motor reference frame [73,74], as well as in integrating arbitrary visual cues to convert it into motor programs [75]. The vPM, connected to the SMA, M1 and the dPM [8,76], plays a role in grasping since it is also connected with the inferior parietal lobule where objects properties are encoded and then converted in an appropriate motor scheme [43,77,78]. Taken together, these findings highlight the critical role of the connectivity between premotor areas and the parietal lobe. These connections are supported by the superior longitudinal system which can be divided into three major branches, i.e., the SLF I, II and III [79], and might explain postoperative movement disorders in the event of damage outside the classical premotor areas [22].

Consequently, glioma resection in the parietal lobe also requires intraoperative movement assessment in awake patients to look for possible motor disturbance, especially at the level of the parietal white matter: DES might elicit NMR at the level of the superior parietal lobule, by stimulating the SLF I or U fibers connecting the postcentral to the precentral gyrus [80]. In addition, a recent report has shown the possibility to elicit movement disturbance by stimulating the anterior intraparietal cortex, which is connected to the premotor area via the SLF II [81]. Finally, DES at the level of the ventral parieto-frontal region might also elicit trouble during the hand manipulation task [22]. Of note, the low number of reports describing NMR at the parietal level is likely due to the low frequency of glioma in this lobe. More studies are mandatory to better describe the cortical and subcortical organization of the network involved in motor control in the parietal lobe.

## 4. The Most Integrated Level: To Action and Conation

Interestingly, patients experiencing NMR related an impossibility to perform the movement, although they perfectly understood the task to do and wanted to do it [47]. They did not report a sudden will to stop movement in the form of a conscious decision to inhibit it. This means that the conation, namely, the willingness which leads to action, was preserved. The patients still experienced the sense of movement sequences, the feeling of making things happen in the sense that they did not want the limb to stop moving, so they did not feel quite rightly responsible for this interruption of action [82]. Fornia et al. also reported that some DES over the precentral gyrus might generate NMR without awareness of the arrest, thus disrupting their sense of agency [83]. Consequently, it happened as if the willingness to act was “disconnected” from the action and agency itself. Such dissociation might be explained by a recent model concerning conation. Indeed, it has been proposed that the anterior insula and habenula are connected in a large network determining the willingness-to-act: remarkably, this network is connected to the SMA which would be responsible for the decision to act or not [84].

This emphasizes one pivotal role of the SMA in movement inhibition [62,85,86,87]. Indeed, Nachev [32] proposed that the SMA relies on a condition-association model where subnetworks inside the SMA could be activated under specific conditions which could then initiate/withhold a movement or elaborate hierarchical rules to manage complex behavior. He supposed that one of the main effects of pre-SMA is mediated by inhibition which is used to suppress competitive programs or action with negative feedback during the rule generation. This might explain the numerous roles attributed to the SMA, as mentioned previously, and seems relevant according to the troubles occurring after lesion or resection in the SMA, far beyond the SMA syndrome [18,31,62]. For example, some patients might present difficulties to inhibit action and, on the contrary, might have a shorter reaction time during the task, in which inhibition usually negatively interferes [85,87]. This can be considered as a conation dysfunction where the willingness to act is too strong and cannot be held, which might favor or interfere with the behavior when talking about reaction time. However, this view remains too hierarchical with a classical organization from intention to action. Indeed, the premotor cortex, and especially the dPM, might play a role in the process of decision making as well as in the state of the motor system, after the impulsion and during the act itself [88,89].

However, why are so few deficits, especially in conation, observed at the ecological level? First, such networks related to conation should be integrated in the meta-networking theory of cerebral function recently proposed [90]. In this framework, the bilateral distribution of the network underlying conation explains the potential of postlesional plasticity, where ipsi- and/or contralateral action can be driven by a unilateral structure [33]. The limitation of this potential may also explain some of the disorders related to specific demands (i.e., bimanual, fast contralateral reaction time when the bilateral/ipsilateral network is mandatory). Thus, on one hand, inter-networks dynamics are a source of resilience by spreading out the load on others network. On the other hand, some functions rely on a more specific circuit (for example, interhemispheric communication for bilateral movement), which represents a limit in functional compensation in case of injury. Second, the consequences of such postoperative deficits, even if subtle, are usually not evaluated in daily life: in particular, they are not correlated with the return to normal activities, such as return to work, which is nonetheless a critical outcome in glioma patients [91]. It is therefore impossible to claim that these slight deteriorations have no ecological impact, depending on the personal definition of quality of life by the patient him/herself, i.e., according to his/her lifestyle, such as practicing sport or art. This is the reason why awake mapping with intraoperative movement monitoring should be adapted in its complexity/sensitivity to the expectations of each patient, after a comprehensive explanation of the risk to induce (or not) an immediate transitory SMA syndrome or more fine but possibly permanent disorders of conation with potential negative consequences on daily activities. In other words, the goal in glioma patients is to propose an individualized medicine centered on patient choices [24,57,57].

In this state of mind, looking for NMR constitutes only the second level of DES neural disruption [27]. Movement is usually considered by neurosurgeons as a specific and single motor task (explaining a simple MEP monitoring in the operating theater), while action and conation in real life should be conceived in terms of permanent on-line monitoring of the best decisions to make that allow the adapted behavior, relying on interplay across several neural networks [90]. Some of these circuits manage high levels goals and are balanced by other circuits managing feedback: remarkably, the state of the motor system itself might influence the decision. Meanwhile, on-line identification of new choices relies on attention and visuo-spatial networks [89,92,93,94,95]. It is worth noting that this metanetwork organization is partly subserved by dense frontoparietal connections through the superior longitudinal system: their critical role pleads for their identification during surgery [79,94,95,96]. In this general framework, performing an action in daily life usually consists of accomplishing motor, cognitive and emotional tasks simultaneously. To those ends, an inter-system coordination is needed thanks to multimodal hubs [27].

Altering such hubs by DES might desynchronize interactions between functional networks and generate difficulties in multitasking (third level) (Figure 4) [27,90]. To prevent persistent deterioration in this higher integrated functions, the patient is asked to perform motor and cognitive task(s) simultaneously with a time constraint, depending on the neural networks surrounding the tumor [23]. This third level of neural disruption by DES mapping requires to conceive movement not in isolation, but as a complex behavioral function necessitating interplay with other cognitive circuits. Therefore, beyond the fact that surgery must be stopped when one specific task cannot be performed (e.g., incapacity to move during DES), resection should also be interrupted when several tasks cannot be performed simultaneously—even if each of them can be performed separately (e.g., incapacity to move and to name while movement alone is still possible or naming is alone is still possible) [23]. To sum up, coordination between tasks become even more important than the task itself, since it is mirroring the adapted functioning of multiple neural networks, a critical integrated process to allow a normal behavior. In a way, this is the ultimate point of conation, where the willingness to act drives the action in its larger sense.

## 5. How to Tailor Resection

As mentioned, resection of glioma might lead to different degrees of impairments depending on the level of movement/conation mapped and preserved during surgery. Consequently, this should be taken into account in terms of onco-functional balance and patient’s information [97]. Choosing to preserve the first level of movement implies simply preserving the motor output. This consists of “only” avoiding hemiplegia/severe hemiparesis by means of MEPs/SEPs under general anesthesia for tumors involving or located near the motor areas in patients who do not need to preserve a high level of complex movement/action/conation.

Choosing to preserve the second level implies monitoring the movement during surgery and therefore performing the resection in awake conditions, regardless of the hemisphere. This management enables us to remove glioma with large extent of resection without impairing the patient’s daily activities, especially the return to work [98]: this is a major point, considering the fact that the patient might be followed after surgery during many years or even decades, especially in low-grade glioma [91,99].

Preserving the third level might be reserved to patients demanding the highest level of conation, i.e., those who need great skill in movement, for example, athletes, musicians, surgeons, etc. Mapping and monitoring the inter-networks dynamics during awake surgery independently of the tumor side is critical for those patients, who might be able to decide to choose a lesser extent of resection to preserve high level control of action in this period of life. Indeed, leaving more residual glioma during the first surgery to avoid (even subtle) functional deteriorations do not prevent reoperation later, after further mechanisms of neuroplasticity have occurred in the meantime: such a multistage surgical approach allows us to increase the extent of resection at reoperation and thus the overall survival, while preserving a high level of quality of life over a period of years [100]. In fact, the reorganization of the network underlying conation may play a role in further resection, according to the kinetics and the pattern of glioma regrowth: thus, the timing of reoperation will depend on the dynamic interaction between neural reconfiguration and the course of tumor relapse at the individual level.

Of course, such examples only imperfectly reflect the complexity and heterogeneity of the daily activities which might help patients to find fulfilment in the context of brain tumor disease. That is why all these elements need to be discussed with the patient and his/her relatives before surgery, to help them to take their best decision in a setting of personalized medicine [24,101].

These decisions rely on an accurate preoperative assessment of conation to identify potential disorders of the first, second or third level, which will lead to tailoring the surgical strategy, by also taking into account the expectations of the patients. Indeed, as first- and second-level troubles can be identified easily on the condition that a fine clinical examination is performed, a possible deficit of the third level is harder to detect, especially because it is patient dependent. This requires us to question the patient concerning his/her daily activities (such as leisure, work, and so forth) in addition to the neurocognitive examination per se. Intraoperative mapping and monitoring will be adapted accordingly, in particular, by personalizing the multitasking used throughout resection. Furthermore, postoperative assessment is also critical to plan an individualized re-habilitation and work adaptation, with the ultimate aim being for the patient to resume a normal life according to his/her wishes.

## 6. Conclusions and Future Directions

The evaluation of motor function during surgery has shown dramatic changes through the century and notably so in the past decade. Such advances have been made possible because the concept of motor function has itself evolved from muscle contraction to conation. This progress has helped us to identify and more precisely predict the consequences of brain damage on movement and action, as well as to adapt the surgical strategy by means of awake DES mapping to prevent functional complications. Thanks to a better representation of the networks subserving such a complex conative function (despite for a long time being considered as “basic”), it is now possible, in a daily neurosurgical practice, to better inform the patient about the putative consequences of glioma resection and to plan the surgery according to his/her expectation in terms of onco-functional balance (awake/asleep, level of conation to preserve).

A deeper understanding and more efficient evaluation of such networks and their inter-dynamics in clinical routine would be helpful to better predict their functional reorganization before surgery and, thus, their capacity of recovery: this would also allow us to ultimately achieve the maximal extent of resection (possibly in a multistage surgical strategy) while preserving the quality of life desired by the patient.

Such an evolution of the concept of a more integrated motor function pleads for a systematic assessment of conation pre- and postoperatively, as well as intraoperatively in awake patients to better explore its meta-networks organization and to identify potential fine but disabling impacts at the ecological level—i.e., regarding the relationships between a human being and his environment, when patients are in the context of their real life. This would improve patients’ information and help them to find the optimal compromise in a permanent virtuous circle based upon stronger interactions between fundamental neurosciences and neurosurgical applications.

## Figures and Tables

**Figure 1 cancers-15-01528-f001:**
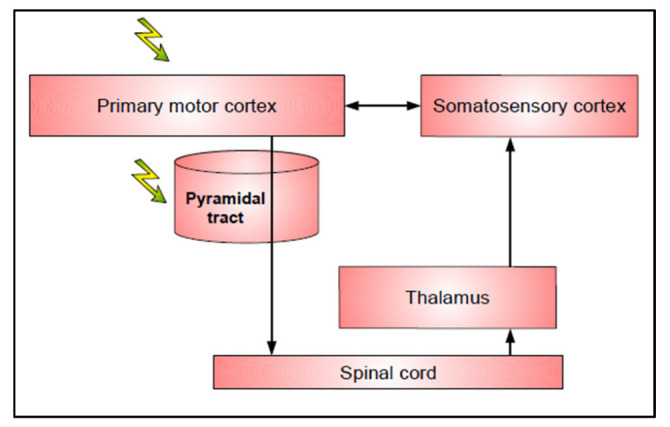
First level of neural disruption by DES: motor output. Flashes represent sites eliciting positive motor responses with DES.

**Figure 2 cancers-15-01528-f002:**
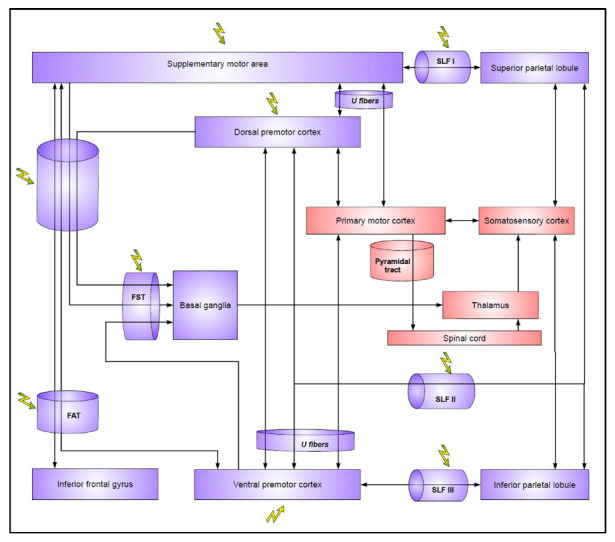
Second level of neural disruption by DES: motor control network. Flashes represent sites eliciting negative motor responses with DES. SLF: superior longitudinal fascicule; FST: fronto-striatal tract; FAT: frontal aslant tract.

**Figure 3 cancers-15-01528-f003:**
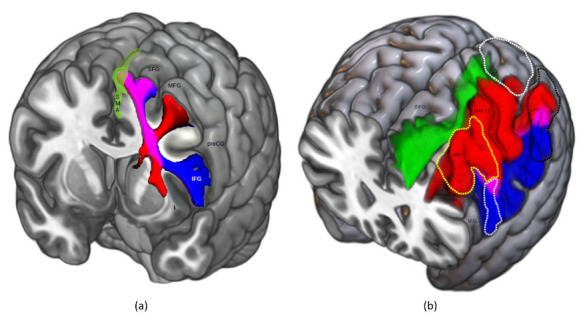
Anatomical relationships between cortical and subcortical structures involved in the second level. (**a**) FAT (blue) and FST (red) trajectories and their cortical projections. The purple zone indicates where these two tracts intermingle. Green area corresponds to the SMA; (**b**) Trajectories and projections of the SLF I (green), SLF II (red) and SLF III (blue). SLF II and III cortical terminations overlapping are show in purple. Posterior part of the dPM (yellow dotted line), vPM (light blue dotted line), superior parietal lobule (white dotted line) and inferior parietal lobule (black dotted line) are also projected to show the relationships between cortical terminations of the SLF I, II and III and fronto-parietal areas involved in motor control. FAT: frontal aslant tract; FST: fronto-striatal tract; SMA: supplementary motor area; SFG: superior frontal gyrus; MFG: middle frontal gyrus; IFG: inferior frontal gyrus; preCG: precentral gyrus; dPM: dorsal premotor cortex; vPM: ventral premotor cortex; SLF: superior longitudinal fascicle.

**Figure 4 cancers-15-01528-f004:**
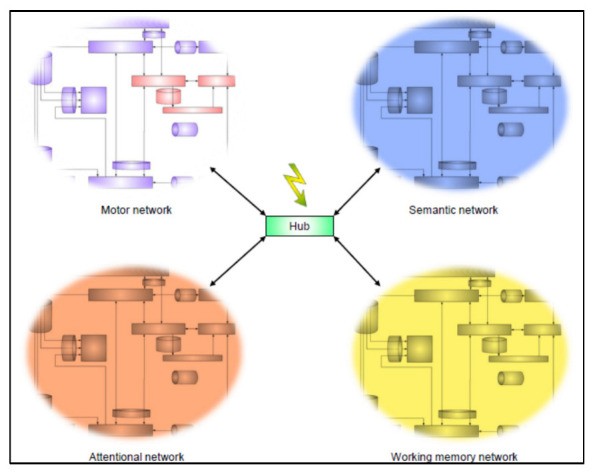
Third level of neural disruption by DES: multitasking. This diagram shows the interplay across the motor network and other neural circuits, in order to illustrate the fact that the system of conation is only part of a large meta-network. Flash represents inter-networks dis-synchronization by stimulating multimodal hub, such as the dorsolateral prefrontal cortex.

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
