# Peer review of "Beyond Avoiding Hemiplegia after Glioma Surgery: The Need to Map Complex Movement in Awake Patient to Preserve Conation"

_cancers, 2023, doi:10.3390/cancers15051528_

Round 1

Reviewer 1 Report

This review is well structured and gives a very nice overview of the insights of Drs Rech and Duffau from their extensive work on motor mapping. The concept of organising the motor system to three orders is a compelling model and might be very relevant for daily practice. The examples of the use of the proposed model in surgical practice are very useful.

I think this a great and informative paper and would totally agree with accepting is as is. Two minor points I would like to make:

Apart from the insights gained inpart 1-4, I think 5 (how to tailor resection) is the most important part of this paper. Maybe the authors can elaborate a bit more on the personalized approach that is necessary in these cases, e.g. by adding a case example (what would be the monitoring and resection approach for a low grade glioma in the dorsal part of the MFG and pre-motor cortex on the non dominant side depending on the level of preservation to be achieved?).

Secondly I would suggest to add a figure detailing the cortical areas and the subcortical tracts referred to in the text, for this might help readers that are less familiar with all pathways discussed.

Author Response

As suggested by the Reviewer, we have added a new figure (figure 3) which brings more insights into the anatomical relationships between the cortical-subcortical structures involved in complex movement referred in the text (FAT / FST/ SLF and cortical fronto-parietal areas): this could be helpful to plan a resection approach into the contact of the pathways underpinning conation.  

Reviewer 2 Report

The manuscript by Dr. Rech and Duffau is interesting, well organized and useful. I have only a minor suggestion to the title, perhaps the first line is unnecessary “Beyond avoiding hemiplegia”.  Also, the term “conation” does not lead to action, only to attempt, other most common terms (movement volition) could be used.

Author Response

We would prefer to let the title unchanged as it introduces the 3 levels mentioned in the work (hemiplegia, complex movement, conation). That being said, we have added a simple summary as demanded by the editor and we have introduced the term “movement volition” suggested by the Referee. This term is also mentioned in the Abstract.

Reviewer 3 Report

The Review by Rech & Duffau presents a summary of comprehensive approaches to monitor complexity of motor function during glioma surgery. Such approach lead to better quality of life for the patients but also to better planning of the surgical resection, taking into account all known risks and gains. The paper states a new level in a field of IOM.

I would just like to suggest reconsidering the following points:

1. In the Abstract (line 29) the Authors mention the importance of pre-surgical conation assesment, but then I really couldn't find any relation in the text to that point. That should be better stated - that's actually one of the things that differentiate Authors' approach from regular neurosurgical monitoring.

2. Regarding point 1 - what methods would be best? electric/magnetic stimulation?

3. In summary / at the end of the article - maybe the block scheme / generalized protocol of the authors approach (pre-intra-post operative steps) would be of value for the reader.

4. The authors mention briefly re-resection - would postoperative assessment of cortical reorganization play a role in that? What would be the timing of such assessment?

5. It would be more visualized for the reader to implement some connectiomic - MRI tractography pictures showing some described anatomical relations.

6. I do not quite understand the use of the word "ecological" in the line 275.

7.Figure 3 needs some upgrading - the network schemes in the circles are the same and have no meaning, the coloured clircles would give the same information. Are there no connection between the circles? Only through the hub?

8. One thorough reading should be made to correct some punctuation/spelling/grammatical/styllistic mistakes.

Author Response

Points 1, 2 and 3: Our main goal is not to propose the use of a presurgical mapping by means of electric/magnetic stimulation, but to emphasize the need of an accurate preoperative assessment of conation, in order to adapt the surgical strategy according to the results of this high level examination as well as to the expectations of the patient – especially with regard to the second and third levels, usually neglected in clinical practice. To this end, we have added a paragraph in part 5 to insist on the importance of such pre-, intra- and post-surgical evaluation to identify a possible second or third level disorder (which is patient-dependent).

Page 11: “These decisions rely on an accurate preoperative assessment of conation to identify potential disorders of the first, second or third level, which will lead to tailor the surgical strategy, by taking also into account the expectations of the patients. Indeed, as first and second level troubles can be identified easily on conditions that a fine clinical examination is performed, a possible deficit of the third level is harder to detect, especially because it is patient-dependent. This requires to question the patient concerning his/her daily activities (such as leisure, work, and so forth) in addition to the neurocognitive examination per se. Intraoperative mapping and monitoring will be adapted accordingly, in particular by personalizing the multitasking used throughout resection. Furthermore, postoperative assessment is also critical to plan an individualized re-habilitation and work adaptation, with the ultimate aim for the patient to resume a normal life according to his/her wishes.”

Point 4: It has been added in the text (page 10) that: “In fact, reorganization of the network underlying conation may play a role in further resection, according to the kinetics and the pattern of glioma regrowth: thus, the timing of reoperation will depend on the dynamic interaction between neural reconfiguration and the course of tumor relapse at the individual level.”

Point 5: As suggested by the Reviewer, we have added a new figure (figure 3) which brings more insights into the anatomical relationships between the cortical-subcortical structures involved in complex movement referred in the text (FAT / FST/ SLF and cortical fronto-parietal areas): this could be helpful to plan a resection approach into the contact of the pathways underpinning conation.  

Point 6: Ecology must be taken in the sense of the relationships between a human being and his environment, that is, when patients are in the context of their real life. This has been added in the main text.

Point 7: The purpose of this diagram is to show the interplay across the motor network and other neural circuits, in order to illustrate the fact that the system of cognation is only part of a large meta-network: this has been added in the Figure legend. The Figure has also  been modified according to the revisions asked by the Editor. 

Point 8: The manuscript has been corrected by an English native speaker.